# PRUNING DEEP CONVOLUTIONAL NEURAL NETWORK USING CONDITIONAL MUTUAL INFORMATION

## ABSTRACT

Convolutional Neural Networks (CNNs) achieve high performance in image classification tasks but are challenging to deploy on resource-limited hardware due to their large model sizes. To address this issue, we leverage Mutual Information, a metric that provides valuable insights into how deep learning models retain and process information through measuring the shared information between input features or output labels and network layers. In this study, we propose a structured filter-pruning approach for CNNs that identifies and selectively retains the most informative features in each layer. Our approach successively evaluates each layer by ranking the importance of its feature maps based on Conditional Mutual Information (CMI) values, computed using a matrix-based Rényi $\alpha$-order entropy numerical method. We propose several formulations of CMI to capture correlation among features across different layers. We then develop various strategies to determine the cutoff point for CMI values to prune unimportant features. This approach allows parallel pruning in both forward and backward directions and significantly reduces model size while preserving accuracy. Tested on the VGG16 architecture with the CIFAR-10 dataset, the proposed method reduces the number of filters by more than a third, with only a $0.32\%$ drop in test accuracy.

## 1 INTRODUCTION

Convolution Neural Network (CNN) has achieved remarkable success in various tasks such as image classification, object detection, and segmentation (Zhang et al., 2019), (Li et al., 2021). Deeper architectures such as VGG16 (Simonyan & Zisserman, 2014) and ResNet (He et al., 2016) have shown superior performance in handling complex image classification tasks. However, the effectiveness of these networks is often reliant on very deep and wide architectures, resulting in a very large number of parameters that lead to longer training and inference time, and create challenges when deploying them on resource-constrained devices (Blalock et al., 2020), (Yang et al., 2017).

CNNs often contain redundant weights and parameters, as certain weights learned in a network are correlated (Sainath et al., 2013). To reduce network size and improve inference speed, network pruning techniques target different components such as weights, filters, and channels, using a range of criteria (see Related Work). A common approach is to measure the weight magnitudes to identify unimportant connections (Han et al., 2015), (Molchanov et al., 2016), (Aghasi et al., 2020).

A less explored approach involves using mutual information between the network's output and latent features to detect redundant filters. Yu et al. (2020) assessed the information flow in CNNs by leveraging the Rényi $\alpha$-order entropy and conducted a preliminary analysis using Conditional Mutual Information (CMI) to identify key filters. However, their study only uses CMI within a single layer, without considering the shared information among features across layers. Furthermore, the CMI-permutation method used to retain filters drastically underestimates the number of useful features. We confirmed in our experiments that the retained features in Yu et al. (2020) lead to a significant drop, of more than $80\%$, in model accuracy.

In this paper, we build upon the concept of using CMI from Yu et al. (2020) to develop an effective method for pruning CNNs while preserving high accuracy. Our key contributions include advancing CMI computation across layers, defining optimal CMI cutoffs, and developing pruning strategies applicable to all CNN layers. Specifically, we introduce novel CMI formulations that capture shared

information across multiple layers, improving the measure's effectiveness in assessing feature importance. We also propose two methods for determining the CMI cutoff point to ensure optimal feature retention. Finally, we develop a robust algorithm for pruning CNN layers bidirectionally, starting from the most critical layer. Evaluations on the VGG16 architecture with the CIFAR-10 dataset demonstrate a 26.83% reduction in parameters and a 36.15% reduction in filters, with only a minimal 0.32% drop in test accuracy, underscoring the effectiveness of our approach. We provide our code and data at: `https://github.com/cmiprune/cmiprune.git`.

## 2 RELATED WORK

Deep neural network pruning has seen major advancements in recent years, with various approaches on reducing model complexity while maintaining performance. These approaches can be categorized into pruning at initialization, dynamic pruning, unstructured pruning, and structured pruning.

*Pruning at initialization* involves selecting weights or neurons likely to contribute little to the overall network performance and removing them without using any gradient steps. Sadasivan et al. (2022) designed OSSuM for pruning at initialization by applying a subspace minimization technique to determine which parameters can be pruned. Tanaka et al. (2020) proposed an approach to measure parameter importance called *synaptic saliency* and ensured that this metric is preserved across layers. However, Frankle et al. (2020) critically examined popular pruning methods at initialization and argued that pruning during training remains more effective.

*Dynamic pruning* approaches adjust the pruning process during training or inference. Shneider et al. (2023) explored disentangled representations using the Beta-VAE framework, which enhances pruning by selectively eliminating irrelevant information in classification tasks. Chen et al. (2023) introduced OTOv3 that integrates pruning and erasing operations by leveraging automated search space generation and solving a novel sparse optimization.

*Unstructured pruning* removes individual weights rather than entire structures like filters, resulting in more flexibility but less hardware efficiency. Molchanov et al. (2019) proposed a Taylor expansion-based pruning method that estimates the importance of weights by their impact on the loss function. Aghasi et al. (2020) introduced Net-Trim, which removes individual weights by formulating the pruning problem as a convex optimization to minimize the sum of absolute entries of the weight matrices. Ding et al. (2019) introduced Global Sparse Momentum SGD, a weight pruning technique that dynamically adjusts the gradient flow during training to achieve high compression ratios while maintaining model accuracy. Lee et al. (2019) demonstrated the role of dynamical isometry in ensuring effective pruning across various architectures without prior training. Han et al. (2015) combined weight pruning, quantization, and Huffman coding to achieve significant compression.

*Structured Pruning* focuses on removing entire channels, filters, or layers, making it more compatible with modern hardware. He & Xiao (2023) provided a comprehensive survey in structured pruning of deep convolutional neural networks, emphasizing the distinction between structured and unstructured pruning and highlighting the hardware-friendly advantages of structured approaches. Crowley et al. (2018) suggested that networks pruned and retrained from scratch achieve better accuracy and inference speed than pruned-and-tuned models. You et al. (2019) developed the Gate Decorator method that employs a channel-wise scaling mechanism to selectively prune filters based on their estimated impact on the loss function, measured through a Taylor expansion. Lin et al. (2022) grouped consecutive output kernels for pruning. Xu et al. (2019) integrated low-rank approximation into the training process, dynamically reducing the rank of weight matrices to compress the network. Considering Convolutional Neural Networks, various approaches have been introduced for filter pruning. Guo et al. (2020) pruned filters using a differentiable Markov process to optimize performance under computational constraints; Sehwag et al. (2020) pruned filters based on an empirical risk minimization formulation; Liu et al. (2019) utilized a meta-learning approach; Molchanov et al. (2016) interleaved greedy criteria-based pruning with fine-tuning by backpropagation, using a criterion based on Taylor expansion to minimize impact on the loss function. Li et al. (2020) developed EagleEye, a pruning method that leverages adaptive batch normalization to quickly and efficiently evaluate the potential of pruned sub-nets without extensive fine-tuning. He et al. (2017) proposed a channel pruning method based on LASSO regression and least squares reconstruction. Zhuang et al. (2018) incorporated additional discrimination-aware losses to maintain the discriminative power of intermediate layers. He et al. (2019) proposed filter pruning via Geometric Median targeting re-

dundant filters to reduce computational complexity. Yu et al. (2020) proposed applying Conditional Mutual Information and Permutation-test to retain a set of important filters.

This paper shares a common objective with prior work in the *structured pruning* domain, particularly focusing on filter pruning for Convolutional Neural Networks. While existing methods employ various pruning criteria, our study explores the application of mutual information (MI), specifically leveraging the matrix-based $\alpha$-order Rényi entropy computation to produce MI values which are used to guide the pruning process. This paper contributes to the area of applying MI in machine learning, emphasizing the use of MI to identify and retain the most informative filters across layers.

## 3   Computing the CMI Values of Candidate Feature Sets

In this section, we analyze the use of Conditional Mutual Information (CMI) as a metric to measure feature importance, and discuss several approaches to ordering the features in each CNN layer and computing their CMI values. We propose new CMI computation that leverages shared information across layers and further exploit Markovity between layers to make the computation efficient.

### 3.1   Selected Features Set and Non-selected Features Set

We first define the notation used for the rest of the paper. Let $X$ and $Y$ be the input and output data of the CNN. We consider a pretrained CNN model that has $N$ CNN layers, $\{L_i\}_{i=1,...,N}$. Each layer $L_i$ contains multiple feature maps obtained by feed-forwarding the training data to this layer using the layer filters. At each layer $L_k$, the feature map selection process involves separating the set of feature maps $F_k$ at layer $L_k$ into two distinct sets: the selected set $F_k^s$ and the non-selected set $F_k^n$, that is, $F_k = \{F_k^s, F_k^n\}$.

**Selected feature set** $F_k^s$ is a subset of the feature map set $F_k$ at layer $L_k$ and consists of feature maps selected according to a selection criterion as discussed later in Section 4. The selection criteria are designed to retain a high test accuracy on the retrained CNN model after pruning.

**Non-selected feature set** $F_k^n$ is the rest of the feature maps at layer $L_k$, i.e. $F_k^n = F_k \setminus F_k^s$, which consists of feature maps that do not significantly contribute to the model's performance, and hence can be pruned to simplify the model complexity without compromising accuracy.

**Selection metric:** We are interested in the information that the feature maps in each layer convey about the CNN output, which can be measured by the mutual information (MI) between the feature map set $F_k$ and the output $Y$. Note the following MI relationship:

$$I(Y; F_k) = I(Y; F_k^s, F_k^n) = I(Y; F_k^s) + I(Y; F_k^n | F_k^s) \tag{1}$$

We observe that the selected feature set $F_k^s$ will convey most information about the output $Y$ if the second term of the summation in Eq. (1) is sufficiently small. This second term measures the conditional mutual information (CMI) between the non-selected feature set and the output, conditioned on the selected feature set. That is to say, *given the selected feature set $F_k^s$, if the non-selected feature set $F_k^n$ does not bring much more information about the CNN output, then it can be effectively pruned without affecting CNN accuracy performance.* As such, in our algorithms, we will compute the CMI values of various candidate feature sets for pruning to determine the best set to prune.

### 3.2   Ordering Features With Per-layer Conditional Mutual Information

We now discuss how to use conditional mutual information (CMI) to rank the feature maps in each CNN layer. The ordered list based on CMI values will later be used for pruning. Here we review the method for ordering features and computing CMI values within one layer as in (Yu et al., 2020); in the next section, we propose new methods for ordering features and computing CMIs across layers.

**Ordering features per layer:** Consider layer $L_k$ with the set of feature maps $F_k$ in a pre-trained CNN. To order the feature maps in $F_k$, we compute the MI between each unordered feature map and the output $Y$, then incrementally select the one that maximizes the MI. Specifically, starting from an empty list of ordered features $F_k^o = [\emptyset]$ and a full list of non-ordered features $F_k^u = F_k$, we successively pick the next best feature map $f^\star$ from $F_k^u$ that maximizes (Yu et al., 2020)

$$f^\star = \underset{f \in F_k^u}{\operatorname{argmax}} \, I(Y; F_k^o \cup \{f\}). \tag{2}$$

Once the next best feature map $f^\star$ is identified, it is moved from the unordered feature list $F_k^u$ to the ordered feature list $F_k^o$ as follows.

$$F_k^o = F_k^o \cup \{f^\star\}; \;\; F_k^u = F_k^u \setminus \{f^\star\}. \tag{3}$$

This process is repeated iteratively for $|F_k|$ times to order all the feature maps of layer $L_k$.

**Computing the per-layer CMI values:** Each time the two lists are updated with a newly ordered feature map as in Eq. (3), they create new candidates for feature selection, where $F_k^o$ is a candidate for the selected feature set, and $F_k^u$ for the non-selected feature set. To evaluate the "goodness" of these candidate sets, we compute the CMI at each ordering iteration $i$ as follows (Yu et al., 2020).

$$c_i = I(Y; F_{k,i}^u | F_{k,i}^o), \quad i = 1 \ldots |F_k| \tag{4}$$

where index $i$ refers to the $i$-th iteration of performing ordering steps (2) and (3) in layer $L_k$.

As $i$ increases, the ordered feature list $F_{k,i}^o$ grows and the non-order feature list $F_{k,i}^u$ shrinks, hence the value of $c_i$ is automatically decreasing with $i$. At the end of this process, each CNN layer will have an associated list of decreasing CMI values $C_k = \{c_1, c_2, \ldots, c_{n_k}\}$, where $n_k = |F_k|$.

### 3.3 Ordering Features With Cross-layer Conditional Mutual Information

The above per-layer CMI computation ignores shared information among features across different layers. To utilize this cross-layer relation, we consider cross-layer CMI computations that incorporate information from multiple CNN layers into the pruning process of each layer. We propose two methods for ordering the features of each layer and computing the cross-layer CMI values.

#### 3.3.1 Full CMI conditioned on all previously considered layers

We follow a similar process as above but replace the maximization criterion in (2) with (5), and the CMI computation in (4) with (6) below. Specifically, let $F_1^s, F_2^s, \ldots, F_{k-1}^s$ be the lists of selected feature maps of previously explored CNN layers $L_1, \ldots, L_{k-1}$. At layer $L_k$, the next feature $f^\star$ to be added to the ordered list $F_k^o$ will be chosen as

$$f^\star = \underset{f \in F_k^u}{\operatorname{argmax}} \, I(Y; F_1^s, \ldots, F_{k-1}^s, F_k^o \cup \{f\}) \tag{5}$$

After updating the ordered list with the new feature map $f^\star$ as in Eq. (3), we calculate the CMI value of the new unordered set as

$$c = I(Y; F_k^u | F_1^s, \ldots, F_{k-1}^s, F_k^o) \tag{6}$$

Steps (5), (3), and (6) are repeated $|F_k|$ times for each layer $L_k$. At the end of this process, each layer again has a list $C_k$ of decreasing CMI values.

#### 3.3.2 Compact CMI conditioned on only the last layer

In feedforward Deep Neural Networks inference, input signals are propagated forward from the input layer to the output layer, passing through multiple hidden layers. In each propagation, the computation flows in a single direction, with the latent features at each layer depending only on the signals from the previously considered layer and weights of the current layer, hence forming a Markov chain (Yu & Principe, 2019). The Markov property implies that the CMI values computed at a certain layer depend solely on the immediately preceding or succeeding layer (Cover, 1999). We stress that this Markov property applies in both directions for CMI computation, whether the given sets that are being conditioned on come from the preceding layers or succeeding layers. (This is because of the property that if $X \to L \to Y$ forms a Markov chain, then $Y \to L \to X$ also forms a Markov chain.) We will later exploit this property to design pruning algorithms that work in both directions. For the easy of exposition, however, we will only show the forward CMI computation here, but noting that it can be applied in the backward direction as well.

Leveraging the Markovity among layers, we propose a more compact method for computing cross-layer CMI values at each layer $L_k$. This method replaces steps (5) and (6) with (7) and (8) respectively as below. The feature ordering maximization criterion becomes

$$f^\star = \underset{f \in F_k^u}{\operatorname{argmax}} \, I(Y; F_{k-1}^s, F_k^o \cup \{f\}) \tag{7}$$

---

**Algorithm 1** Feature ordering with CMI Computation

---

1: **Input:** Selected features set $F_1^s, F_1^s, \ldots, F_{k-1}^s$ of layer $L_1, L_2, \ldots, L_{k-1}$, full feature set of current layer $F_k$, output $Y$
2: **Initialize:** $F_k^o = [\emptyset], F_k^u = F_k, C_k = [\emptyset]$
3: **while** $|F_k^u| \geq 1$ **do**
4:     Find $f^\star$ according to Eq. (2) or Eq. (5) or Eq. (7)
5:     Update: $F_k^u = F_k^u \setminus \{f^\star\}; \quad F_k^o = F_k^o \cup \{f^\star\}$
6:     Compute CMI value $c$ according to Eq. (4) or Eq. (6) or Eq. (8), respectively as in Step 4
7:     Append $C_k = \{C_k, c\}$
8: **end while**
9: **return** $F_k^o, C_k$

---

and the compact CMI computation used to create the CMI list is

$$c = I(Y; F_k^u | F_{k-1}^s, F_k^o) \tag{8}$$

Steps (7), (3), and (8) are repeated $|F_k|$ times for each layer $L_k$ to produce the CMI list $C_k$.

### 3.3.3 FULL CMI VERSUS COMPACT CMI AND EXAMPLES

While the compact CMI in (8) and the full CMI in (6) are theoretically equivalent because of Markovity among CNN layers, their numerical values may vary in practice due to the estimation methods used for calculating mutual information and the numerical precision of the machine. Specifically, we use the matrix-based numerical method for computing Rényi entropy in (**??**) (see Appendix) from layer data without having the true distributions, thus the computed values for compact CMI and full CMI diverge when conditioned on more layers. Therefore, we conduct an ablation study to compare both approaches in the experimental evaluation presented in Section 6.

Algorithm 1 provides the implementation details of feature ordering and CMI computation for all three methods: per-layer CMI, cross-layer full CMI, and cross-layer compact CMI. The algorithm returns the fully ordered feature set $F_k^o$ of layer $L_k$ and the set of decreasing CMI values $C_k$.

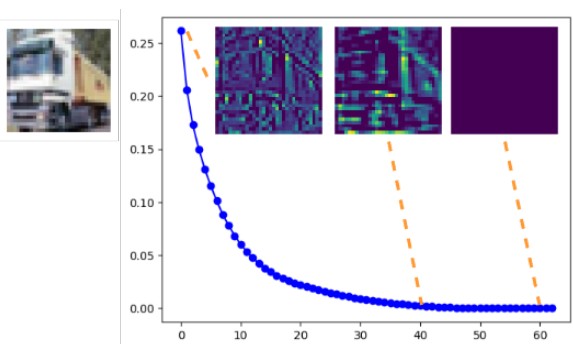

Figure 1 provides an example illustrating the ordered feature maps in a CNN layer based on cross-layer compact CMI values. This particular CNN layer has 64 feature maps, whose indices are shown on the horizontal axis in the order of decreasing CMI values as shown on the vertical axis. At index points 1, 40, and 60, we display the corresponding newly added feature map to the ordered feature set. The first feature map shows a relatively clear pattern related to the input image of a *truck*, while the middle one becomes more blurry, and the last feature map does not at all resemble the truck. In the next section, we

Figure 1: Example of ordered feature maps using cross-layer compact CMI computation in Alg. 1. The top left figure is the input image with label *truck*. The vertical axis presents the computed CMI value and the horizontal axis shows the index of the newly added ordered feature map.

present two different approaches, Scree test and X-Mean clustering, for selecting a cutoff point to prune the feature maps based on CMI values. Using these approaches, the added feature maps at points 1 and 40 are retained, whereas the feature map at point 60 is consistently pruned. This means the set of last five feature maps from 60 to 64 contains little information about the CNN output and can be pruned without affecting accuracy performance.

---

**Algorithm 2** Determining CMI Cutoff Point in a Layer Using Scree Test

---

1: **Input:** List of ordered features $F_k^o$ from layer $L_k$, list of CMI values $C_k$, pre-trained CNN model $M$, training dataset $D$, target accuracy threshold $a^p$, number of top candidates $K$
2: **Initialize:** List of cut-off index and model accuracy $A = [\ ]$
3: **for** $i = 1$ to $(|C_k|-2)$ **do**
4:     $s(i) = \frac{c_i - c_{i+1}}{c_{i+1} - c_{i+2}}$    // *Compute QDA score for each CMI point*
5: **end for**
6: **Find** top $k$ largest $s(i)$ values and their corresponding indices $\{i_1, i_2, \ldots, i_K\}$
7: **for** $j = 1$ to $K$ **do**
8:     Prune all features in $F_k^o$ with indices after $i_j$ to obtain an intermediate pruned model $M_j$
9:     Evaluate $M_j$ on $D$ to obtain the accuracy $a_j$
10:     Append $(i_j, a_j)$ to $A$
11: **end for**
12: Choose the smallest index $i^\star$ with $a_{i^\star} \geq a^p$ or else $i^\star = \max\{i_1, \ldots, i_K\}$
13: Select all features up to index $i^\star$, and prune all features after $i^\star$ in $F_k^o$, to obtain $F_k^s$
14: **return** $(i^\star, F_k^s)$

---

## 4   DETERMINING A CUTOFF POINT FOR CMI VALUES IN EACH LAYER

After ordering the features of each CNN layer and computing the CMI values of candidate sets of features as in Section 3, the features are arranged in descending order of CMI values. The next step is to determine a cutoff point within the ordered list of CMI values such that the set of features with CMI value at the cutoff point is selected and retained, and the set of features with lower CMI, which contributes little to the CNN output, is pruned. In this section, we propose two methods to identify such a cutoff point based on the Scree test and X-Mean clustering.

### 4.1   IDENTIFYING CUTOFF POINT USING SCREE TEST

The Scree test (Cattell, 1966) is first proposed in Principal component analysis (PCA) to determine the number of components to be retained using their eigenvalues plotting against their component numbers in descending order. The point where the plot shifts from a steep slope to a more gradual one indicates the meaningful component, distinct from random error (D'agostino Sr & Russell, 2005). Furthermore, Niesing (1997) introduced the Quotient of Differences in Additional values (QDA) method, which identifies the $q^{th}$ component that maximizes the slope $s(q) = (\lambda_q - \lambda_{q+1})(\lambda_{q+1} - \lambda_{q+2})^{-1}$ where $\lambda_q$ is the eigenvalue for the $q^{th}$ component in PCA.

Here we apply the QDA method (Niesing, 1997) to the list of decreasing CMI values obtained as in Section 3. To explore more than one candidate cutoff point, we propose to find $K$ CMI values that correspond to the top $K$ largest slopes as

$$\{i_1, i_2, \ldots, i_K\} = \underset{i=1\ldots|F_k|-2}{\text{top K}} \frac{c_i - c_{i+1}}{c_{i+1} - c_{i+2}}, \tag{9}$$

Each of the $K$ candidate cutoff points from the list obtained above will be examined by carrying out trial pruning of current layer $L_k$ (pruning off the set of features beyond each point) and testing the resulting pruned model for accuracy. (This pruned model is the one obtained right at this pruning step in the current layer and is not the final pruned model.) The optimal cutoff point will then be chosen based on the resulting pruned model's accuracy while maximizing the pruning percentage. Specifically, denote $a^f$, $a^p$ as the accuracy of the full and pruned models, respectively, and let $\delta$ be the targeted maximum reduction in accuracy such that $a^f - a^p \leq \delta$. Then the optimal cutoff point is the one from (9) which results in the largest pruned percentage while satisfying the accuracy requirement. If no candidate point meets this accuracy threshold, the index with the highest accuracy is chosen. Since this process involves trial pruning and testing for accuracy of the pruned model, typically only a small value of $K$ is used, around 2 or 3 cutoff point candidates. In the special case of $K = 1$, only the cutoff point with maximum slope is chosen and no trial pruning is necessary. Algorithm 2 outlines the procedure for selecting the optimal cutoff point using the Scree test.

---

**Algorithm 3** Determining CMI Cutoff Point in a Layer using X-Means Clustering

---

1: **Input:** List of ordered feature maps $F_k^o$ of layer $L_k$, list C of CMI values, pre-trained model $M$, training dataset $D$, accuracy threshold $a^p$
2: Apply X-means on $C$ to obtain $K$ clusters $e_1, \ldots, e_K$, ordered in the decreasing CMI value of the cluster center
3: **Initialize**: A = [ ], $F_k^s = F_k^o$
4: **for** $j = 1$ to $K$ **do**
5:      Append features in $e_j$ to A
6:      Prune features in $e_{j+1}$ to $e_K$ from model $M$ to obtain an intermediate pruned model $M_j$
7:      Evaluate $M_j$ on $D$ to obtain accuracy $a_j$
8:      **If** $a_j >= a^p$ **then** $F_k^s \leftarrow$ A and **break**
9: **end for**
10: **return** $F_k^s$

---

## 4.2 IDENTIFYING CUTOFF POINT USING X-MEANS CLUSTERING

Here we propose an alternative method to select the optimal CMI cutoff point based on clustering using the X-Means algorithm (Pelleg et al., 2000), an extension of $k$-means, to cluster the CMI values into different groups. X-Means automatically determine the optimal number of clusters based on the Bayesian Information Criterion $\text{BIC}(M) = \mathcal{L}(D) - \frac{p}{2}\log(R)$ where $\mathcal{L}(D)$ is the log-likelihood of dataset $D$ with $R$ samples according to model $M$ with $p$ parameters.

X-Means starts with an initial cluster number, and increases this number until the BIC score stops improving. Once clusters are formed in the current layer, we order the clusters based on the CMI value of the cluster center point in decreasing order. Starting with the first cluster, we retain all its feature maps and perform trial pruning of the remaining feature maps from all other clusters. The pruned model's accuracy is then evaluated. As the process continues, new features from the next cluster are added to the selected feature set, until the test accuracy meets or exceeds the targeted accuracy threshold. Algorithm 3 provides the outline of this X-Means procedure.

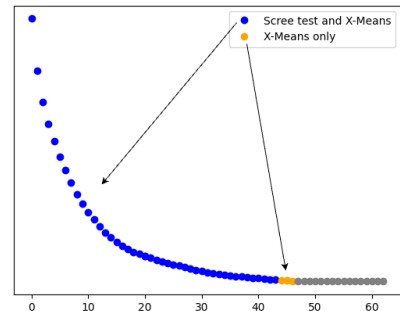

Figure 2: Example of cutoff points by Scree test and X-Means.

Figure 2 illustrates the cutoff points selected by using the Scree test and X-Means clustering methods. We see that the majority of feature maps selected by the Scree test and X-Means clustering are similar, represented by the blue points. The orange points indicate feature maps retained only by X-Means, and the gray points represent feature maps pruned by both methods. The difference between the two methods boils down to only the last few feature maps. In this example, the Scree test retains 43 while X-Means retains 46 out of the total 64 feature maps.

## 5 ALGORITHMS FOR PRUNING ALL LAYERS OF A CNN BASED ON CMI

We now combine methods from the previous two sections in an overall process to systematically traverse and prune every layer of a CNN. We propose two algorithms that differ in their starting layer and pruning direction. One algorithm begins at the first convolutional layer and prunes forward through the network. The other starts at the layer with the highest per-layer pruning percentage and simultaneously prunes both forward and backward from there.

The pruning process consists of three phases as illustrated in Figure 3. The first phase is *Data Preparation* which generates the feature maps of each layer. We start with a pre-trained CNN model that feeds forward the data using mini-batch processing through each CNN layer $L_k$ to produce a set of feature maps $F_k$. The second stage is the main *Pruning Algorithm* in which every convolutional layer of the CNN is processed and pruned in a certain order. The last stage is *Retraining* of the pruned model to fine-tune the model parameters to improve accuracy performance.

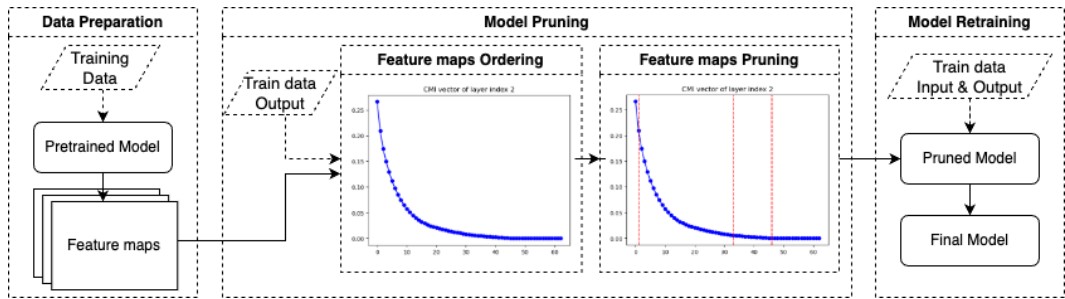

Figure 3: Overview of the CMI-based pruning process. The blue curve shows a list of decreasing CMI values as new feature maps are sequentially added to the order set of each layer. The red vertical lines indicate candidate cutoff points for the CMI list. The important feature maps to be selected and retained are those to the left of the red lines.

---

**Algorithm 4** Forward Pruning Procedure

---

    **Input:** Set of feature maps $\{F_1, F_2, \ldots, F_N\}$, output $Y$, pre-trained CNN model $M$, model accuracy $a^f$, accuracy threshold $a^p$, training data $D$

1: **for** $k = 1$ to $N$ **do**
2:     $C_k, F_k^o \leftarrow$ Rank features in $L_k$ using *cross-layer CMI* (Alg. 1) with inputs $F_{k-1}^s, F_k, Y$
3:     $F_k^s \leftarrow$ Find cutoff point within *CMI list* (Alg. 2 or 3) with inputs $C_k, F_k^o, M, D, a^p$
4: **end for**
5: **return** selected feature set for each layer $F_1^s, \ldots, F_N^s$

---

### 5.1 FORWARD MODEL PRUNING

In *Forward Pruning*, the algorithm starts with the first convolutional layer and prunes all convolutional layers sequentially from first to last. At each layer, the algorithm applies the chosen feature ordering and CMI computation method (Section 3) to produce the decreasing CMI value list, then applies the chosen cutoff point identification method (Section 4). In cross-layer CMI computation, the CMI values of each layer are computed by conditioning on the selected feature sets of previous layers. Algorithm 4 describes this forward pruning procedure.

### 5.2 BI-DIRECTIONAL MODEL PRUNING

We design *Bi-directional Pruning* to improve the previous pruning approach by first determining the most effective layer to begin the pruning process. We propose to start with the layer that has the highest per-layer pruning percentage while maintaining an acceptable post-pruning accuracy. First, we perform trial-pruning of each convolutional layer of the CNN individually, using *per-layer CMI computation* and either the Scree test or X-Means method. This initial stage lets us identify the layer with the highest pruning percentage as the starting layer for the full CNN pruning process. Next, we start from the identified best layer and proceed by using *cross-layer CMI computation* to prune the original CNN in both directions, forward and backward. For *compact CMI* computation, at each new layer, the compact CMI values are conditioned on the immediately previous layer that was pruned, which can either be the preceding layer (in forward pruning) or the succeeding layer (in backward pruning). For *full CMI* computation, we condition the CMI on all previously pruned layers from the starting layer in the corresponding direction. We note that in Bi-directional pruning, per-layer CMI computation in Eq. (4) is only used at the initial stage to determine the starting layer; after that, the pruning process uses cross-layer CMI computation in Eq. (8) or Eq. (6). Algorithm 5 outlines the detailed procedure of Bi-directional Pruning.

## 6 EXPERIMENTAL RESULTS

This section presents our experimental evaluation of the CNN pruning algorithms. Due to space, we present the main results here and delegate detailed results and ablation studies to the Appendix.

---

**Algorithm 5** Bi-directional Pruning Procedure

---

**Input:** Set of feature maps $\{F_1, F_2, \ldots, F_N\}$, output $Y$, pre-trained model $M$, accuracy of pre-trained model $a^f$, accuracy threshold $a^p$, training data $D$

1: **for** $k = 1$ to $N$ **do**
2:   $C_k, F_k^o \leftarrow$ Compute *per-layer CMI* (Alg. 1)
3:   $F_k^s, a_k \leftarrow$ *Prune* $(C_k, F_k^o)$ *using Scree test or X-Means* (Alg. 2 or 3)
4:   $r_k \leftarrow 1 - |F_k^s|/|F_k|$   *// pruning ratio $r_k$*
5: **end for**
6: Determine layer $k^\star$ with the highest pruning percentage $r_{k^\star}$ and $a_{k^\star} >= a^p$
    *// Forward CMI Computation*
7: **for** $k = k^\star + 1$ **to** $N$ **do**
8:   $C_k, F_k^o \leftarrow$ Compute *cross-layer CMI* values (Alg. 1) for layer $L_k$
9:   $F_k^s \leftarrow$ *Prune* $(C_k, F_k^o)$ *using Scree test or X-Means* (Alg. 2 or 3)
10: **end for**
    *// Backward CMI Computation*
11: **for** $k = k^\star$ - 1 **down to** 1 **do**
12:   $C_k, F_k^o \leftarrow$ Compute *cross-layer CMI* values (Alg. 1) for layer $L_k$
13:   $F_k^s \leftarrow$ *Prune* $(C_k, F_k^o)$ *using Scree test or X-Means* (Alg. 2 or 3)
14: **end for**
15: **return** Set of selected features for each layer $F_1^s, \ldots, F_N^s$

---

## 6.1 EXPERIMENT SETUP

We evaluate our proposed pruning algorithms on VGGNet (Simonyan & Zisserman, 2014), specifically a VGG16 model which consists of 13 convolutional layers (Phan, 2021), pre-trained on the CIFAR-10 dataset (Krizhevsky et al., 2009). We use the training data to evaluate the accuracy of the intermediate pruned models, and the test data to evaluate the accuracy of the final pruned model. When preparing the data, we use a batch of 256 training samples to feed forward through the VGG16 model and generate the feature maps at each layer for use in our algorithms.

We performed several experiments to prune the original CNN model using different combinations of CMI computation and cutoff point methods as in Algorithms 4 and 5. When using the Scree-test with multiple candidates, we set $K = 3$. The original accuracy on training data is $99.95\%$ (Phan, 2021), and to check the accuracy of the intermediately pruned models, we set the target accuracy as $a^p = 98.95\%$. In all experiments in this section, *we prune the CNN model by completely removing the weights corresponding to the pruned features in each layer (Actual pruning – see Appendix)*. The final convolutional layer is not pruned to maintain all connections to the first fully connected layer. The pruning efficiency is determined by the percentage of pruned filters over all filters.

After the CNN model is fully pruned, we *re-train each pruned model* to fine-tune the weights for better test accuracy. For the retraining process, we apply the VGG16 training parameters for CIFAR-10 as in (Phan, 2021) and train each pruned model with 100 epochs.

## 6.2 ANALYSIS OF FEATURE MAPS ORDERING AND CMI COMPUTATION METHODS

Table 1 shows a comparative analysis of the various feature maps ordering and CMI computation approaches as discussed in Section 3 (Algorithms 1). The cutoff point selection method in this set of experiments is the Scree-test. The results are displayed in terms of the number of retained parameters, pruned percentage of filters, and test accuracies before and after retraining.

The Bi-directional pruning algorithm with cross-layer compact CMI computation (Algorithm 1) yields the smallest pruned model size (24.618 M parameters retained), representing $26.84\%$ parameter reduction from the original model. The same algorithm also results in the highest *pruned percentage* of $36.15\%$ filters removed. Although this most aggressive pruning approach leads to a slightly lower accuracy *before retraining* compared to other approaches, it actually achieved the best test accuracy *after retraining*. After retraining, all considered methods converged to a similar accuracy. The original model's test accuracy was $94\%$, and after retraining for 100 epochs, this most aggressively pruned model achieves a test accuracy of $93.68\%$, which is the best among all exper-

Table 1: CNN Pruning using Scree-test Cutoff Point with various CMI Computation Methods

| CNN Pruning Algorithms | Parameters Retained | Filters Pruned Percentage | Accuracy before Retraining | Accuracy after Retraining |
|---|---|---|---|---|
| *No pruning (original model)* | *33.647 M* | *0 %* | *94.00%* | – |
| Forward pruning & full CMI | 33.196 M | 2.18% | 93.02% | 93.67% |
| Forward pruning & compact CMI | 25.7 M | 26.70% | 90.17% | 93.33% |
| Bi-directional pruning & full CMI | 25.643 M | 30.12% | 88.59% | 93.25% |
| Bi-directional pruning & compact CMI | **24.618 M** | **36.15%** | 90.95% | **93.68%** |

Table 2: Bi-directional Pruning with Compact CMI using Various Cutoff Point Approaches

| Cutoff Point Approaches | Parameters Retained | Filters Pruned Percentage | Accuracy before Retraining | Accuracy after Retraining |
|---|---|---|---|---|
| *No pruning (original model)* | 33.647 M | 0 % | 94.00% | - |
| Permutation-test (Yu et al., 2021) | 19.379 M | 81.79% | 9.99% | 10.02% |
| Scree-test | **24.618 M** | **36.15%** | **90.95%** | **93.68%** |
| X-means | 25.01 M | 34.67% | 83.56% | 92.99% |

imented methods. *This result confirms the validity of our approach of using cross-layer compact CMI computation and pruning in both directions.*

### 6.3 ANALYSIS OF CMI CUTOFF POINT APPROACHES

In this set of experiments, we compare the two proposed CMI cutoff point approaches, Scree-test and X-means, with the Permutation-test in (Yu et al., 2021). For Permutation-test, we use a permutation number of 100 and a significance level of 0.05 as used in (Yu et al., 2021). The CNN pruning algorithm is Bi-directional Pruning with Cross-layer Compact CMI computation (Alg. 5). Table 1 shows the effectiveness of different cutoff point approaches when applied to the VGG16 model.

The Permutation-test (Yu et al., 2021) shows the smallest pruned model size but at a drastically reduced test accuracy to only 10.02% even after retraining. This shows that the Permutation test was not able to differentiate unimportant features from the important ones and hence pruned aggressively and indiscriminately. In contrast, the proposed Scree-test and X-means both achieve more than a third of the features pruned while still retaining most of the accuracy of the original model. The results show that Scree-test is slightly more robust than X-means by achieving both a higher pruned percentage and a better retrained-accuracy. This could be because Scree-test is more effective at preserving the most important feature maps compared to X-means.

## 7 CONCLUSION

In this study, we introduced novel structured pruning algorithms for Convolutional Neural Networks (CNNs) by using Conditional Mutual Information (CMI) to rank and prune feature maps. By applying matrix-based Rényi $\alpha$-order entropy computation, we proposed several CMI-based methods for identifying and retaining the most informative features while removing redundant ones. Two different strategies, Scree test and X-means clusterng, were explored to determine the optimal cutoff points for pruning. We also examine both forward and backward prunings which were found to be effective. Our experiments demonstrated that the proposed approach significantly reduces the number of parameters by more than a third with negligible loss in accuracy, achieving efficient model compression. This method provides a promising framework for deploying CNN models on resource-constrained hardware without compromising performance. Future work may explore extending this approach to other network architectures and tasks beyond image classification.

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
