# A APPENDIX

## A.1 BACKGROUND

### A.1.1 CONVOLUTIONAL NEURAL NETWORKS

Convolutional Neural Networks (CNN) is a specialized type of deep neural network primarily used for processing structured grid-like data such as images (Younesi et al., 2024). CNN is particularly effective in image processing tasks such as image classification or object detection, because of its ability to automatically learn and extract *hierarchical features* from the input data. Different CNN architectures have been introduced for image processing tasks, including LeNet (LeCun et al., 1998), AlexNet (Krizhevsky et al., 2012), Visual Geometry Group (VGG) (Simonyan & Zisserman, 2014), Residual Network (ResNet) (He et al., 2016) and MobileNet (Howard, 2017).

A CNN architecture generally consists of an input layer, a stack of alternating convolutional and pooling layers, several fully connected layers, and an output layer at the end (Zhao et al., 2024). The top panel in Fig. 4 shows the VGG-16 architecture, which includes 13 convolutional layers and 3 fully connected layers. Each convolutional layer contains a set of filters. A convolution operation involves sliding a filter over the input image, multiplying the filter values by the pixel values at corresponding positions in the input image, and summing the results to obtain a feature map. By applying various filters to the input image, a set of feature maps is generated, as shown in Fig. 4. When multiple convolutional layers are stacked, the later layers capture more representative features of the input image. We will use the VGG-16 architecture as the main example for implementation in this paper, but all the discussion and developed algorithms can be applied to any CNN structure.

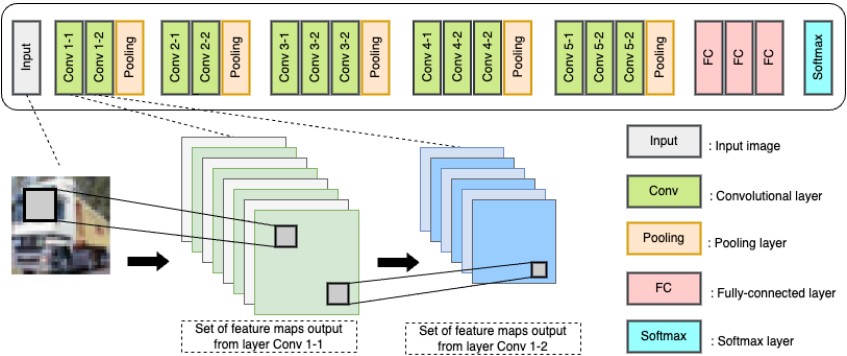

Figure 4: Illustration of the process of a sample CNN model.

### A.1.2 MULTIVARIATE MUTUAL INFORMATION USING RÉNYI ENTROPY

Our proposed CNN pruning method is based on computing the conditional mutual information between the features extracted in the same layer and in different layers of the CNN. Each feature is treated as a multivariate random variable in matrix form. The test data after being processed through the trained CNN provides samples or realizations of each random feature at each layer. Next, we discuss the method used for computing the mutual information (MI) and conditional mutual information (CMI) subsequently.

**Rényi Entropy and Mutual Information Computation:** To estimate MI between random variables, we rely on the Rényi's $\alpha$-order entropy $H_\alpha(X)$ (Rényi, 1965), defined as

$$H_\alpha(X) = \frac{1}{1-\alpha} \log \left( \int_X p^\alpha(x) \, dx \right), \tag{10}$$

where $X$ is a continuous random variable with the probability density function (PDF) $p(x)$, and $\alpha$ is a positive constant. Rényi entropy extends the well-known Shannon entropy which is obtained when the parameter $\alpha$ approaches 1 (Rényi, 1965).

Calculating Rényi entropy requires knowing the PDF, which limits its application in data-driven context. To overcome this, we employ a matrix-based $\alpha$-order Rényi entropy calculation (Giraldo

---

**Algorithm 6** CMI permutation test (Yu & Principe, 2019a)

---

1: **Input:** Selected ordered set of feature maps $F_k^s$, remaining feature maps $F_k^r$, class labels $Y$, selected feature map $f$ (in $F_k^r$), permutation number $P$, significance level $\alpha$
2: **Compute:** Estimate $I(\{F_k^r - f\}; Y \mid \{F_k^s, f\})$
3: **for** $i = 1$ to $P$ **do**
4:     Randomly permute $f$ to obtain $\tilde{f}_i$
5:     Estimate $I(\{F_k^r - \tilde{f}_i\}; Y \mid \{F_k^s, \tilde{f}_i\})$
6: **end for**
7: *Evaluate the significance:*
8: **if** $\frac{1}{P} \sum_{i=1}^{P} \mathbf{1}[I(\{F_k^r - f\}; Y \mid \{F_k^s, f\}) \geq I(\{F_k^r - \tilde{f}_i\}; Y \mid \{F_k^s, \tilde{f}_i\})] \leq \alpha$ **then**
9:     $F_k^s \leftarrow F_k^s \cup f$
10:     $\texttt{decision} \leftarrow$ Continue feature map selection
11: **else**
12:     $\texttt{decision} \leftarrow$ Stop feature map selection
13:     $N \leftarrow |F_k^s|$
14: **end if**
15: **return** $\texttt{decision}, N$

---

et al., 2014) which computes Rényi's $\alpha$-order entropy using the eigenspectrum of a normalized Hermitian matrix, derived by projecting data into a Reproducing Kernel Hilbert Space (RKHS) (Gong et al., 2022):

$$S_\alpha(G) = \frac{1}{1-\alpha} \log_2 \left( \text{tr}(G^\alpha) \right) = \frac{1}{1-\alpha} \log_2 \left( \sum_{i=1}^{n} \lambda_i^\alpha(G) \right), \tag{11}$$

where $G$ is a normalized kernel matrix obtained from the data and $\lambda_i(G)$ are the eigenvalues of $G$.

For a given CNN, to construct matrix $G$, we first extract latent features from the CNN by feed-forwarding the training data to each CNN layer. This process provides for each layer a feature matrix $\mathbf{X}^{N \times d}$, where each row represents a $d$-dimensional feature vector of a data sample. We then compute the kernel matrix $\hat{G}$ from these features using a kernel function $\varphi(x_i, x_j)$ that measures the similarity between feature vectors $x_i$ and $x_j$. In our experiment, we use the RBF kernel $\varphi(x_i, x_j) = \exp(-||x_i - x_j||^2 / (2\sigma^2))$. Next, we normalize the kernel matrix $\hat{G}$ to obtain the normalized kernel matrix $G$. The normalization ensures $G$ is symmetric and its eigenvalues are within the range $[0, 1]$.

For multiple variables, the matrix-based Rényi's $\alpha$-order joint entropy of $L$ variables is computed as (Yu et al., 2019)

$$S_\alpha(G_1, G_2, \ldots, G_L) = S_\alpha \left( \frac{G_1 \circ G_2 \circ \cdots \circ G_L}{\text{tr}(G_1 \circ G_2 \circ \cdots \circ G_L)} \right), \tag{12}$$

where $(G^k)_{ij} = \varphi_k(x_i^k, x_j^k)$, with $k \in \{1, ..., L\}$ denotes the normalized kernel matrix of the $\text{k}^{th}$ variable, and $\varphi_k \colon \mathcal{X}^k \times \mathcal{X}^k \mapsto \mathbb{R}$ is the $\text{k}^{th}$ positive definite kernel, and $\circ$ denotes the Hadamard product.

Using Rényi entropy, the matrix-based Rényi's $\alpha$-order mutual information $I_\alpha(\cdot; \cdot)$ is computed as

$$I_\alpha(G; G_1, \ldots, G_L) = S_\alpha(G) + S_\alpha(G_1, \ldots, G_L) - S_\alpha(G_1, \ldots, G_L, G) \tag{13}$$

**Conditional Mutual Information Computation using Rényi Entropy:** Conditional mutual information (CMI) quantifies the amount of information shared between two random variables, $X$ and $Y$, given the knowledge of a third variable $Z$. Typically, it is expressed using Shannon entropy as

$$I(X; Y|Z) = H(X, Z) + H(Y, Z) - H(X, Y, Z) - H(Z) \tag{14}$$

Using Rényi entropy, CMI can be generalized as the matrix-based Rényi $\alpha$-order CMI:

$$I_\alpha(G_X; G_Y|G_Z) = S_\alpha(G_X, G_Z) + S_\alpha(G_Y, G_Z) - S_\alpha(G_X, G_Y, G_Z) - S_\alpha(G_Z), \tag{15}$$

where $G_X, G_Y, G_Z$ are the normalized kernel matrices defined on the data samples of the variables $X, Y$, and $Z$, respectively.

Table 3: Comparison of Permutation Test, Scree Test, and X-Means on *Individual Layer pruning* with per-layer CMI. Each test accuracy value is shown for the pruned model obtained by pruning *only* the current layer. Accuracy values above 90% are in **bold**.

| | | PERMUTATION TEST | | SCREE TEST | | X-MEANS | |
| Layer No. | Total #Filters | #Filters Selected | Acc. | #Filters Selected | Acc. | #Filters Selected | Acc. |
|---|---|---|---|---|---|---|---|
| 1 | 64 | 2 | 12.83% | 49 | **94.00%** | 47 | **94.00%** |
| 2 | 64 | 2 | 9.99% | 60 | **92.89%** | 47 | **91.27%** |
| 3 | 128 | 2 | 10.00% | 124 | **93.40%** | 111 | **93.16%** |
| 4 | 256 | 8 | 8.40% | 109 | **91.91%** | 111 | **92.39%** |
| 5 | 256 | 2 | 9.99% | 229 | **93.17%** | 223 | **92.45%** |
| 6 | 256 | 1 | 9.99% | 247 | **93.44%** | 239 | **92.48%** |
| 7 | 512 | 19 | 20.95% | 238 | **93.71%** | 159 | **91.71%** |
| 8 | 512 | 17 | 10.23% | 414 | **93.68%** | 265 | **92.58%** |
| 9 | 512 | 23 | 80.63% | 218 | **93.13%** | 244 | **93.58%** |
| 10 | 512 | 19 | **93.97%** | 192 | **93.71%** | 140 | **93.62%** |
| 11 | 512 | 19 | **94.00%** | 215 | **93.66%** | 195 | **93.59%** |
| 12 | 512 | 79 | **94.00%** | 326 | **94.02%** | 136 | **93.79%** |
| 13 | 512 | 359 | **93.78%** | 448 | **93.92%** | 51 | **93.53%** |

## A.2 PERMUTATION TEST

We describe in this section the *Permutation Test* used by (Yu & Principe, 2019a) to quantify the impact of a new feature map $f$ on the model accuracy. Specifically, for a new feature $f$, CMI permutation test creates a random permutation $\tilde{f}$ from $\{f \cup F_k^s\}$, and computes the new CMI value between the output $Y$ and the set of unselected features, conditioned on the permutation set $\tilde{f}$. The algorithm then compares this new CMI value with the original CMI that is conditioned on the original set $\{f \cup F_k^s\}$ to determine whether the contribution of feature $f$ on the output is significant. Specifically, if the CMI value of the permutated feature set is not significantly smaller than the original CMI value, the permutation test will discard feature $f$, as $f$ does not capture the spatial structure in the input data, and stop the feature selection process. However, applying CMI permutation method on CNN models leads to the retention of very few filters (Yu et al., 2021), resulting in a significant drop in the model accuracy. We describe the CMI permutation test as used for feature selection in (Yu et al., 2021) in Algorithm 6.

## A.3 DIFFERENT CUTOFF POINT APPROACHES ON PER-LAYER CMI

In this section, we compare three approaches, Permutation test, Scree test and X-means, for determining the cutoff point of CMI values and evaluate their effectiveness on *per-layer CMI*. Here we prune each layer individually without pruning any other layers, and evaluate the accuracy performance of the resulting pruned model with one layer pruned. The results are provided in Table 3, showing that the Permutation test retains high accuracy in only 4 out of 13 convolutional layers, while both the Scree test and X-means maintain high accuracy in all layers. The impact of using the Permutation test to prune all layers is even more dramatic as seen by the results in Table 2.

## A.4 FULL CMI VERSUS COMPACT CMI ON FORWARD PRUNING

In this section, we present the experimental results of Forward Pruning in 4 with two methods for ranking features and computing CMI values: *Full CMI* (Section 3.3.1) and *Compact CMI* (Section 3.3.2), using Scree test as the cutoff point method. Table 4 presents the results of the number of selected filters and the corresponding accuracy of the pruned model after iteratively pruning each layer. We observe that, for the first 12 layers, Full CMI retains more filters than Compact CMI and

Table 4: Full CMI versus Compact CMI on Forward Pruning with Scree test, using *Zero weight* pruning where the non-selected filters are set to 0 but not removed from the CNN. Each test accuracy value is shown for the pruned model obtained by pruning all layers from the first layer up to and including the current layer, without retraining.

| Layer No. | Total #Filters | FULL CMI #Filters Selected | Acc. | COMPACT CMI #Filters Selected | Acc. |
|---|---|---|---|---|---|
| 1 | 64 | 49 | 94.00% | 49 | 94.00% |
| 2 | 64 | 59 | 93.55% | 59 | 93.59% |
| 3 | 128 | 124 | 93.48% | 108 | 92.95% |
| 4 | 256 | 125 | 93.47% | 125 | 92.95% |
| 5 | 256 | 252 | 93.26% | 209 | 91.37% |
| 6 | 256 | 252 | 93.04% | 251 | 91.33% |
| 7 | 512 | 248 | 92.95% | 248 | 91.24% |
| 8 | 512 | 504 | 92.93% | 355 | 90.19% |
| 9 | 512 | 505 | 92.93% | 405 | 89.81% |
| 10 | 512 | 501 | 92.95% | 197 | 88.73% |
| 11 | 512 | 507 | 92.95% | 323 | 87.71% |
| 12 | 512 | 505 | 92.95% | 255 | 88.19% |
| 13 | 512 | 11 | 37.79% | 408 | 87.38% |

hence results in a smaller decrease in accuracy. However, in the last CNN layer, Full CMI retains very few filters, leading to the significant drop in the pruned model's accuracy. On the other hand, Compact CMI has a higher pruned percentage by retaining fewer filters in most layers (except the last one) while maintaining relatively consistent accuracy throughout all layers.

## A.5 COMPARISON BETWEEN FEATURES RETAINED BY SCREE TEST AND X-MEANS

To examine in more detail the difference between Scree test and X-means, we analyze the selected feature sets of each approach using Bi-directional pruning with Compact CMI computation. Table 5 shows the comparison. The *Overlap* presents the percentage of feature maps that are retained by both Scree test and X-means, relative to the total number of feature maps in a given layer. This "Overlap" measure provides insight into the agreement between the two cutoff point approaches regarding which feature maps are essential. *Scree test Only* and *X-means Only* represent the percentage of feature maps retained exclusively by the Scree test and X-means, respectively, relative to the total number of features retained by each approach. We can see that the overlap of selected features between the two approaches is highest for Layer 6 and gradually decreases the farther away from this layer. This overlap percentage is in agreement with the percentage of filters pruned shown for each approach, as Layer 6 has the lowest percentage pruned for both methods. We note also that the starting layer for pruning with Scree-test is Layer 10, and with X-means is Layer 13. The percentage of filters pruned is highest for each method at its starting layer and decreases from there, but not necessarily in a strictly decreasing order the farther away from the starting layer. This result is quite curious and shows that different sets of filters can be pruned at each layer depending on the cutoff point method while still preserving the final accuracy within a relatively reasonable range. The final re-trained pruned model obtained with either Scree-test or X-means has a test accuracy within $1.01\%$ of the original unpruned model (as shown in Table 2).

## A.6 ANALYSIS ON PRUNING TYPES: ZERO WEIGHTS VERSUS ACTUAL PRUNING

In this experiment, we consider two types of pruning: *Zero weight*, which sets the pruned weights to zero while keeping the network structure unchanged, and *Actual pruning*, which completely removes the pruned weights from the network, thereby reducing the number of parameters and memory

Table 5: Comparison of Shared and Exclusive retained feature maps between Scree test and X-means on Bi-directional pruning with Compact CMI. The "Overlap" column shows the percentage of overlapping selected filters, and the last two columns show the individual percentage of filters pruned, all relative to the total number of filters in each layer. The "Only" columns show the percentage of uniquely selected filters relative to the total number of selected filters in each method. The star (*) indicates the starting layer for pruning in each method.

| LAYER Index | OVERLAP | SCREE TEST Only | X-MEANS Only | %FILTERS PRUNE | |
| :---: | :---: | :---: | :---: | :---: | :---: |
| | | | | Scree Test | X-Means |
| 1 | 68.75% | **0.00%** | 6.38% | 31.25% | 26.56% |
| 2 | 73.44% | 22.95% | **0.00%** | 4.69% | 26.56% |
| 3 | 86.72% | 10.48% | **0.00%** | 3.13% | 13.28% |
| 4 | 86.72% | 8.26% | **0.00%** | 5.47% | 13.28% |
| 5 | 92.19% | **0.00%** | 1.26% | 7.81% | 6.64% |
| 6 | 93.36% | 4.78% | **0.00%** | 1.95% | 6.64% |
| 7 | 83.20% | 0.47% | 4.48% | 16.41% | 12.89% |
| 8 | 55.86% | 30.07% | **0.00%** | 20.12% | 44.14% |
| 9 | 52.15% | **0.00%** | 44.49% | 47.85% | 6.05% |
| 10 | 26.17% | 30.21% | 4.29% | **62.50 %** (*) | 72.66% |
| 11 | 27.73% | 29.35% | 15.98% | 60.74% | 66.99% |
| 12 | 47.46% | 2.80% | 26.81% | 51.17% | 35.16% |
| 13 | 9.96% | 85.51% | **0.00%** | 31.25% | **90.04%** (*) |

usage. During *Actual pruning*, as we focus on CNN layers, we leave the last CNN layer unpruned to preserve its connections to the following fully connected layer.

These two pruning types also involve a difference in the BatchNorm layer operation following each pruned CNN layer. In Zero-weight pruning, we set the pruned filters to zero without adjusting the BatchNorm layer. In actual pruning, however, the pruned filters are completely removed from the CNN model, hence the shape of each pruned CNN layer changes and we adjust the BatchNorm operation accordingly to match the smaller shape. These adjustments lead to different test accuracies between Zero-weight and Actual pruning for the pruned models.

Table 6 shows the comparison between Zero-weight and Actual pruning with different CNN pruning and CMI computation methods. We use the Scree test for selecting the cutoff point. The results show that *Zero-weight* pruning leads to higher pruned percentage compared to *Actual pruning* for three out of the four settings. However, *Actual pruning* consistently leads to higher test accuracy for the final pruned model across all settings. We also note that Bi-directional pruning with compact CMI achieves the best performance, with highest pruned percentage in both pruning types while still maintaining high accuracy even before re-training.

Finally, Table 7 shows the comparison between *Zero-weight* and *Actual pruning* using different cutoff point methods. The CNN pruning and CMI computation methods are Bi-directional pruning and Compact CMI, respectively. The results show that the *pruned percentage* of Permutation test is highest compared to other cutoff point methods in both pruning types. However, Permutation test results in extremely low accuracy both before and after retraining, making it unsuitable for practical purposes. The Scree test provides highest accuracy among all methods in both pruning types.

Table 6: Zero weight versus Actual pruning using Scree test Cutoff Point with various CMI Computation Approaches and Pruning Directions

| CNN PRUNING | FEATURES ORDERING | PRUNING TYPE | |
|---|---|---|---|
| | | Zero-weight | Actual pruning |
| **Filters Pruned Percentage** | | | |
| Forward pruning | full CMI | 13.78% | 2.18% |
| Forward pruning | compact CMI | 29.17% | 26.70% |
| Bi-directional pruning | full CMI | 34.04% | 30.12% |
| Bi-directional pruning | compact CMI | **35.56%** | **36.15%** |
| **Parameters Retained (unpruned model: 33.647 M)** | | | |
| Forward CMI | full CMI | - | 33.196 M |
| Forward CMI | compact CMI | - | 25.7 M |
| Bi-directional pruning | full CMI | - | 25.643 M |
| Bi-directional pruning | compact CMI | - | **24.618 M** |
| **Accuracy *before Retraining* (unpruned model: 94.00%)** | | | |
| Forward CMI | full CMI | 37.79% | 93.02% |
| Forward CMI | compact CMI | 87.38% | 90.17% |
| Bi-directional pruning | full CMI | 84.95% | 88.59% |
| Bi-directional pruning | compact CMI | **82.12%** | **90.95%** |
| **Accuracy after Retraining** | | | |
| Forward CMI | full CMI | - | 93.67% |
| Forward CMI | compact CMI | - | 93.33% |
| Bi-directional pruning | full CMI | - | 93.25% |
| Bi-directional pruning | compact CMI | - | **93.68%** |

Table 7: Zero weight vs. Actual pruning on Bi-directional Pruning with Compact CMI using Various Cutoff Point Approaches

| CUTOFF POINT METHOD | PRUNING TYPE | |
| --- | --- | --- |
| | **Zero-weight** | **Actual pruning** |
| **Filters Pruned Percentage** | | |
| Permutation test | **75.50%** | **81.79%** |
| Scree test | 35.56% | 31.77% |
| X-mean | 41.38% | 34.67% |
| **Parameters Retained (unpruned model: 33.647 M)** | | |
| Permutation test | - | **19.379 M** |
| Scree test | - | 24.618 M |
| X-means | - | 25.01 M |
| **Accuracy before Retraining (unpruned model: 94.00%)** | | |
| Permutation test | 9.99% | 9.99% |
| Scree test | **82.12%** | **90.95%** |
| X-means | 22.09% | 83.56% |
| **Accuracy after Retraining** | | |
| Permutation test | - | 10.02% |
| Scree test | - | **93.68%** |
| X-means | - | 92.99% |