# OpenReview forum: "Pruning Deep Convolutional Neural Network Using Conditional Mutual Information"
_ICLR.cc/2025/Conference — ICLR 2025 Conference Withdrawn Submission_

### Official Review · Reviewer_e6ih · 2024-11-01

**Soundness:** 2
**Presentation:** 3
**Contribution:** 3
**Rating:** 5
**Confidence:** 4

**Summary:**

This paper proposes a structured pruning method to achieve the compact DNNs. The proposed method leverages the conditional mutual information to demonstrate the importance of filters in each layer. Then, either the Scree test or X-Mean clustering is used to determine the pruning cutoff point for removing unimportant filters. Moreover, two pruning modes (forward pruning and bi-directional pruning) are investigated in this paper.

**Strengths:**

1. Code is released.
2. The paper is generally well-written, and the approach is very comprehensive.
3. The experimental results show the effectiveness.

**Weaknesses:**

1. This paper only conducts experiments on VGG-16 on CIFAR-10, hence the generalization capability of the proposed method seems to be limited. The authors should report the performance on larger datasets (such as ImageNet) and the models with different architectures (such as ResNet).
2. The paper lacks the comparison with other state-of-the-art pruning methods to highlight its effectiveness.
3. The algorithm seems to be quite complex, as each layer requires numerous iterations to calculate CMI and the cutoff point. Additionally, it requires a large memory footprint to store the feature maps from previous layers for calculating cross-layer CMI. Does this imply that the algorithm may be difficult to deploy in compressing deeper networks or those with larger input size?
4. In Algorithm 4, when $k=1$, what does $F ^ s _ {0}$ represent?
5. In the experiment, a batch of 256 training samples is used to measure the filters. Does the number of samples affect the stability of the measurement? Can different samples produce the similar measurements? When more samples are used to calculate the measurement, can a more stable measurement be obtained?
6. Why is the pruning efficiency determined by the percentage of pruned filters over all filters (Line 468 on Page 9)?

\
Minor issues:
1. $L _ i$ and $L_k$ should be unified in the first paragraph of Section 3.1.
2. $F ^ s _ 1$ in Line 1 of Algorithm 1 is repeated.
3. There is a symbol error in Line 241 on Page 5.
4. 'k' should be 'K' in Line 6 of Algorithm 2.
5. The references are repeated from Line 648 to Line 655 on Page 13.

**Questions:**

Please see the weaknesses.

---

### Official Review · Reviewer_nPQA · 2024-11-01

**Soundness:** 1
**Presentation:** 1
**Contribution:** 2
**Rating:** 1
**Confidence:** 5

**Summary:**

This paper proposes a novel approach to structured pruning of Convolutional Neural Networks (CNNs) using Conditional Mutual Information (CMI). The authors present methods for ranking feature importance using CMI, determining pruning cutoff points, and implementing pruning strategies in both forward and bi-directional manners. The work demonstrates large model size reduction with small accuracy loss on VGG16 with CIFAR-10.

**Strengths:**

1. The method part has a lot of information.
2. The metric of mutual information is rather new.

**Weaknesses:**

1. The largest weakness is the experiments. I'm supervised to see that only the old network VGG16 is tested on a small dataset CIFAR-10. Normally, for model pruning, ResNet-50 on ImageNet is necessary. Recently, Vision Transformer is also critical to validate the algorithm. Missing these important experiments makes this paper far below the standard of ICLR.
2. Missing important references. Many recent papers in 2024, such as [a], are not included. The author should do a complete survey.
3. Performance is not good. In [a], there is no accuracy drop when pruning 60+% FLOPs.


[a] Auto-Train-Once: Controller Network Guided Automatic Network Pruning from Scratch

**Questions:**

Please see weakness.

---

### Official Review · Reviewer_F4xf · 2024-11-04

**Soundness:** 2
**Presentation:** 1
**Contribution:** 1
**Rating:** 1
**Confidence:** 5

**Summary:**

The authors propose a structured pruning method to reduce the computational complexity of CNNs. It uses conditional mutual information (CMI) to select feature maps that can be pruned from the network and applies an iterative re-training to compensate for the pruning errors.

**Strengths:**

Although other methods have been proposed to prune DNNs based on the mutual information (e.g. [1],[2]), it might be that the proposed selection criterion has some novelty.

[1] Ganesh, Madan Ravi, Jason J. Corso, and Salimeh Yasaei Sekeh. "Mint: Deep network compression via mutual information-based neuron trimming." 2020 25th International Conference on Pattern Recognition (ICPR). IEEE, 2021.
[2] Hussien, Mostafa, et al. "Small Contributions, Small Networks: Efficient Neural Network Pruning Based on Relative Importance." arXiv preprint arXiv:2410.16151 (2024).

**Weaknesses:**

There are several weaknesses:
* The paper only promises results on CIFAR10. However, this is simply not enough for a compression paper these days.
* The paper seems to be incomplete. Results are promised in the appendix, but the appendix is missing.
* The only results provided in the paper (Table 1 and Table 2) show variants of their own pruning method, but do not compare to other literature.
* all plots miss axis labels
* References are missing (e.g. reference of Renyi entropy)

**Questions:**

* Could you please comment how your method compares to [1]
* Further, I would like to see the results from the appendix


[1] Ganesh, Madan Ravi, Jason J. Corso, and Salimeh Yasaei Sekeh. "Mint: Deep network compression via mutual information-based neuron trimming." 2020 25th International Conference on Pattern Recognition (ICPR). IEEE, 2021.

---

### Note · Authors · 2024-11-14

I have read and agree with the venue's withdrawal policy on behalf of myself and my co-authors.